# *AvNAC030*, a NAC Domain Transcription Factor, Enhances Salt Stress Tolerance in Kiwifruit

**DOI:** 10.3390/ijms222111897

**Published:** 2021-11-02

**Authors:** Ming Li, Zhiyong Wu, Hong Gu, Dawei Cheng, Xizhi Guo, Lan Li, Caiyun Shi, Guoyi Xu, Shichao Gu, Muhammad Abid, Yunpeng Zhong, Xiujuan Qi, Jinyong Chen

**Affiliations:** 1Zhengzhou Fruit Research Institute, Chinese Academy of Agricultural Sciences, Zhengzhou 450009, China; 13598072703@163.com (Z.W.); guhong@caas.cn (H.G.); chengdawei@caas.cn (D.C.); guoxizhi@caas.cn (X.G.); lilan01@caas.cn (L.L.); shicaiyun@caas.cn (C.S.); xuguoyi@caas.cn (G.X.); 82101182230@caas.cn (S.G.); zhongyunpeng@caas.cn (Y.Z.); qixiujuan@caas.cn (X.Q.); chenjinyong@caas.cn (J.C.); 2Lushan Botanical Garden, Chinese Academy of Sciences, Jiujiang 332900, China

**Keywords:** kiwifruit, salt tolerance, oxidative stress, ROS, NAC

## Abstract

Kiwifruit (Actinidia chinensis Planch) is suitable for neutral acid soil. However, soil salinization is increasing in kiwifruit production areas, which has adverse effects on the growth and development of plants, leading to declining yields and quality. Therefore, analyzing the salt tolerance regulation mechanism can provide a theoretical basis for the industrial application and germplasm improvement of kiwifruit. We identified 120 NAC members and divided them into 13 subfamilies according to phylogenetic analysis. Subsequently, we conducted a comprehensive and systematic analysis based on the conserved motifs, key amino acid residues in the NAC domain, expression patterns, and protein interaction network predictions and screened the candidate gene *AvNAC030*. In order to study its function, we adopted the method of heterologous expression in *Arabidopsis*. Compared with the control, the overexpression plants had higher osmotic adjustment ability and improved antioxidant defense mechanism. These results suggest that *AvNAC030* plays a positive role in the salt tolerance regulation mechanism in kiwifruit.

## 1. Introduction

Soil salinization can destroy the ionic and osmotic balance of plant cells, inhibit their growth and development, and reduce the yield and quality of crops, making soil salinization a worldwide problem that restricts the healthy and sustainable development of modern agriculture [1]. In arid agricultural areas, soil salinization is becoming more and more serious due to the lack of rainfall, strong light, and other factors that will lead to the accumulation of soluble salt in the soil on the surface, coupled with improper irrigation and excessive fertilization [2]. For these reasons, over 800 million hectares of land around the world are affected by salt, more than 6% of the world’s total land area [3]. At present, 45 million hectares (19.5%) of 230 million hectares of arable land in the world are affected by soil salinization, and, due to climatic factors and unreasonable irrigation, this number is increasing year by year [4,5,6]. Therefore, soil salinization has become one of the main limiting factors restricting the development of agriculture worldwide [7]. The soil replacement method, trenching and salt drainage, chemical reagent improvement, water and fertilizer regulation, and other measures are common soil improvement methods, but these methods are time-consuming, laborious, and easily lead to soil hardening. Cultivating salt-tolerant crops, as well as the selection of salt-tolerant rootstocks of fruit trees, without excluding the chance to enhance the suitable native wild species, are the most economical, effective, safe, and environmentally friendly methods [8,9]. As one of the four most successful artificially domesticated and cultivated trees in the 20th century, kiwifruit (*Actinidia chinensis* Planch) has a unique flavor and is rich in vitamin C, which is the antiviral vitamin par excellence and used as a cure for COVID-19. Kiwifruit has the effect of clearing the intestine and strengthening the stomach, and is increasingly favored by consumers [10]. Huang et al. believed that *Actinidia* Lindl. has a total of 75 taxa, consisting of 54 species and 21 varieties, which are mainly distributed in China and neighboring countries [11]. Although more than 100 varieties (or strains) have been selected from wild or seeding populations and cross breeding through breeding programs in countries such as China, New Zealand, Italy, and Chile, little research has been conducted on assessing rootstock resistance, especially salt tolerance [12,13]. Kiwifruit is a fleshy root that prefers neutral and acidic soil [14]. Its roots are mostly distributed in the upper soil about 40 cm beneath the surface. This depth is also the area of salt accumulation and deposition [15]. Some major varieties of kiwifruit are sensitive to salt, which seriously affects the fruit yield and quality. The problem of soil salinization in some kiwifruit production areas is becoming increasingly prominent, and salt stress has become the main obstacle to the sustainable development of the kiwifruit industry.

The growth limit salinity of kiwifruit is low. When the soil salinity concentration reaches 0.14%, it will cause salt damage to the plant and interfere with the normal growth and development of the plant. When the salt concentration reaches 0.54%, the kiwifruit yield will decrease sharply and the salinity may even lead to the death of the plant [16]. Salt damage in plants can be attributed to ion stress, osmotic stress, and oxidative stress [17]. After Na^+^ enters the cell, it accelerates the degradation of metabolism-related enzymes in the cytoplasm, reduces the K^+^/Na^+^ ratio, destroys the resting potential of the cell membrane, and leads to metabolic disorder and ion toxicity [18]. The increase in Na^+^ and Cl^−^ content in soil affects the absorption of Fe^2+^ by the roots and leads to leaf yellowing, which is particularly serious in kiwifruit. In addition, the absorption of Ca^2+^, K^+^, HPO_4_^2−^, and NO^3−^ is also affected, resulting in plant ion imbalance. The lack of mineral nutrients obstructs the formation and transport of photosynthates, thus reducing the content of ATP and nutrients, resulting in nutrient deficiency and ion stress. [19]. In salinized soil, the water potential of plant root cells is higher than the external environment, the water potential difference cannot be used to absorb water, and the water absorption amount is lower than the transpiration amount, leading to physiological drought. In severe cases, the water in the plant penetrates outward and causes dehydration, resulting in osmotic stress [20]. After salt stress, photosynthesis is inhibited, a large number of electrons are accumulated, and the content of reactive oxygen species (ROS) increases rapidly, resulting in the degradation of enzymes, nucleic acids, and other macromolecular substances. In addition, ROS can destroy the structure of chloroplasts, mitochondria, and the cell membrane, resulting in oxidative stress [21]. After salt damage, kiwifruit morphology is mainly characterized by the inhibition of plant growth, the decline of organic matter accumulation, an insufficient supply of nutrients, and short branches and internodes [22]. The plant may stop growing or even die in serious cases.

During the long-term evolution of plants, a set of resistance mechanisms against salt stress have been developed [23]. These mechanisms are mainly divided into salt avoidance mechanisms and salt tolerance mechanisms, among which salt tolerance mechanisms are divided into ion balance and regionalization, osmotic adjustment, and antioxidant defense mechanisms [24]. After sensing the external stress, the plasma membrane will trigger the transmission of calcium signals, an SOS pathway, and hormones so as to activate the response mechanism [25]. Transcription factors are the key factors linking salt stress response signals with plant salt tolerance regulatory networks and can precisely regulate downstream target genes [26]. Under salt stress, transcription factors can simultaneously regulate multiple downstream stress-responsive genes, activate the salt-tolerant response of plants, and reduce or eliminate the damage caused by salt stress [23]. At present, the reported transcription factor families related to the mechanism of plant salt tolerance include NAC, MYB, WRKY, bZIP, AP2/ERF, and bHLH [27]. Among them, NAC transcription factors play a key role in regulating the response mechanism of plants to salt stress [28]. Under salt stress, plants can regulate cell osmotic pressure by accumulating osmotic regulators, stabilizing protein and membrane structures, and eliminating ROS through oxidative defense mechanisms to reduce the damage caused by oxidative stress [29]. Wang et al. found that rice plants overexpressing *ThNAC13* improved salt tolerance by accumulating osmotic regulatory substances and scavenging ROS [30]. After overexpression of *OoNAC72* in *Arabidopsis thaliana*, Guan et al. found that transgenic plants carry out osmotic regulation and remove ROS after salt stress so as to reduce peroxidation damage [31]. Li et al. used gene editing combined with genetic transformation and other molecular biology techniques and found that *GmNAC06* could reduce the content of ROS in plants through the accumulation of osmotic mediating substances, thereby increasing the salt tolerance of plants [32].

We screened *Actinidia valvata* germplasm material ZMH (Zhenmu, Hunan) with strong salt tolerance and rootstock application prospects in the early stage [33]. We used this as a material to screen 120 *AvNAC* genes based on conserved domains. Then, we conducted a systematic and comprehensive analysis of the NAC family, including systematic evolutionary relationships, conservative motifs, protein network interaction prediction, and key amino acid residue distribution, and combined this data with sequencing results to screen candidate genes [34,35,36,37,38,39]. Subsequently, we verified the function of the *AvNAC030* gene by heterologous expression in *Arabidopsis*. The phenotypic analysis, molecular experiments, and physiological parameters showed that *AvNAC030* increased plant salt tolerance. The above results have important theoretical and practical significance for further understanding the molecular mechanism of salt tolerance in kiwifruit and accelerating the cultivation of salt-tolerant rootstocks and varieties.

## 2. Results

### 2.1. Phylogenetic Analysis of the NAC Family in Kiwifruit

NAC (NAM, ATAF1,2, and CUC2) protein, as a plant-specific transcription factor, is widely distributed in terrestrial plants [40]. The diversity of NAC family members indicates the diversity of their functions, which are related to plant growth and development and stress responses [41]. Family members with close relatives may have similar functions, so phylogenetic analysis is of guiding significance for gene function prediction [42]. Taking the *Arabidopsis* NAC family as a reference, we used the nomenclature protocol to construct an unrooted phylogenetic tree of 120 NAC members of kiwifruit according to the multiple sequence alignments of conserved domains (Figure 1) [43,44]. On the basis of Heim’s method, we made a few appropriate adjustments (Table 1). For example, the NAC2 subfamily was divided into the VII a and VII b subfamilies. The TERN subfamily and ONAC022 subfamily were merged into the IX subfamily and formed a sister subfamily. Subfamily NAP and subfamily AtNAC3 in subfamily X are also sister subfamilies, implying their co-evolution [45,46]. Finally, according to 105 NAC members of *Arabidopsis*, the kiwifruit NAC family was divided into 13 subfamilies. Subfamily II has no AvNAC members, which may be the result of long-term evolution. *AtNAC097* could not be classified in any of these 14 subfamilies and was therefore classified as an orphan [47]. The number of members in different subfamilies varies greatly. Subfamily VII b and IX, with the largest number of family members, both contain 22 AvNACs, while the subfamilies V and XII, with the smallest number, contain 2 AvNACs. These results provide evidence for the evolutionary relationship of the kiwifruit NAC family.

### 2.2. The Motif Analysis of the NAC Family in Kiwifruit

Motifs play an important role in the interaction of different modules in the signal transduction and transcription complex [48,49]. We analyzed the sequence, length, distribution, and frequency of 20 conserved motifs of 120 VvNAC genes (Figure 2 and Table 2). Motif1, motif2, motif3, motif4, and motif5, which occur frequently, are mainly distributed at the N-terminal region of the NAC domain, indicating that these conservative motifs play an important role in the function of VvNAC [50]. Some less frequent motifs only appear in a specific subfamily. Motif6 appears only in subfamily III. Motif9, motif11, motif12, motif16, motif19, and motif20 also only appear in specific subfamilies, which may be related to the specific functions of these subfamilies. Therefore, both the number and type of motifs in different subfamilies are quite different. The average number of motifs in each subfamily is 3 to 7, and the types are 5 to 11. Each type of motif appears only once in each gene. However, the occurrence times of each motif are different. Motif3 appears 98 times, and motif13 and motif20 appear only 6 times. There are also great differences in the motif types of each gene. Some genes have 10 types, and some have only one type.

### 2.3. Analysis of Conserved Amino Acid Residues in the NAC Domain of Kiwifruit

A typical NAC protein consists of a conserved N-terminal NAC region (about 150 amino acids) and a diverse C-terminal transcriptional regulatory region [51,52]. The NAC domain with DNA binding ability in *NAC* transcription factors can be divided into five subdomains. The highly conserved positively charged C and D subdomains are responsible for binding to DNA. Nuclear localization signals (NLSs) present in C and D subdomains may be related to nuclear localization in transcription factors and the recognition of specific cis-acting elements on promoters. A subdomain is involved in the formation of functional dimers. B and E subdomains are not conservative and are responsible for the functional diversity of NAC genes [53]. In order to better understand the functions of the kiwifruit NAC family, we conducted multiple sequence array analysis of its 120 members (Figure 3A). Subsequently, we compiled statistics on the percentage of conserved amino acids in the five subdomains based on the previous report, and the results showed that there were 14 sites in which the consistency rate exceeded 75%. Among them, the D subdomain with DNA binding ability and containing NLSs contained the most sites, with eight sites. The A subdomain contained three sites, and the C subdomain contained two sites. The non-conserved B domain contained only one site, and the E domain had no sites.

### 2.4. The Expression Level of the AvNAC Family under Salt Stress

It was previously reported that members of the NAC family could improve the salt tolerance of plants [54]. The expression pattern of genes is related to their function [55]. In order to study the function of the NAC family in kiwifruit under salt stress, we used the salt-tolerant resource ZMH as a material to analyze the expression patterns of NAC family members after 0 (I), 6 (II), 24 (III), and 72 (IV) hours of salt stress (Figure 4). The fragments per kilobase per million (FPKM) values were used to estimate the expression characterization of the NAC family for screening the candidate genes associated with salt tolerance. The results showed that the expression of *AvNAC030* and *AvNAC031* of subfamily IV, *AvNAC037* of subfamily VII a, *AvNAC060* of subfamily IX, and *AvNAC098* of subfamily XII increased significantly after salt stress.

### 2.5. The Interaction Network Analysis of Candidate Genes

The prediction of gene interaction networks can help researchers understand gene functions quickly and effectively [56]. Therefore, we used STRING to predict the candidate gene interaction network based on the AvNAC orthologs in *Arabidopsis* (Figure 5). The expression of *AvNAC030* (*NAC019* in *Arabidopsis*) was induced by salt stress, and its interaction gene *RHA2A* was able to respond positively to salt stress and osmotic stress, while ZFHD1 was regulated by salt stress. *AvNAC031* (*NAC041* in *Arabidopsis*) is the transcription activator of the mannan synthase CSLA9. It can recognize and bind to the DNA-specific sequence of the CSLA9 promoter. *AvNAC037* (*NAC100* in *Arabidopsis*) can bind to the promoter regions of genes involved in chlorophyll catabolic processes. *AvNAC060* (*NAC070* in *Arabidopsis*) can control the cell wall maturation processes that are required to detach root cap layers from the root. *AvNAC098* (*NST1* in *Arabidopsis*) is a transcription activator of genes involved in the biosynthesis of secondary walls. Together with NST2 and NST3, *AvNAC098* is required for the secondary cell wall thickening of sclerenchymatous fibers, secondary xylem (tracheary elements), and of the anther endocethium, which is necessary for anther dehiscence. It may also regulate the secondary cell wall lignification of other tissues. Based on the above results, it is speculated that *AvNAC030* may be involved in the regulation mechanism of salt tolerance in kiwifruit.

### 2.6. Subcellular Localization of AvNAC030 

We fused the green fluorescent protein (GFP) to the C-terminus of *AvNAC030* with a mutation in the stop codon, and used the CaMV35S constitutive promoter to drive it to determine its subcellular location. Subsequently, *35S::AvNAC030:GFP* fusion protein and control *35S::GFP* were transferred into *Arabidopsis* protoplasts by a PEG-mediated method (Figure 6). *AtBZR2* was fused to mCherry as a nuclear marker. The *Arabidopsis* protoplasts with *35S::GFP* plasmid displayed fluorescence throughout the cells. In contrast, the *Arabidopsis* protoplasts with *35S::AvNAC030:GFP* plasmid was detected only in the nucleus. This result suggests that *AvNAC030* may encode a nuclear localized protein.

### 2.7. The Effects of Overexpression of AvNAC030 in Arabidopsis

Four-week-old homozygous T_3_-generation *Arabidopsis* were used to study the function of *AvNAC030* in response to salt stress in a substrate treated with 250 mM NaCl solution. The phenotype of overexpression (OE) plants was significantly superior to that of Vector control (VC) plants, although both OE and VC plants were damaged to varying degrees after salt treatment (Figure 7A). OE plants also had a higher survival rate after 4 weeks of salt treatment (Figure 7B). Subsequently, we determined the content of flavonoids with ROS scavenging abilities, and the results showed that the accumulation of total flavonoids in OE plants after salt treatment was significantly higher than that of VC plants (Figure 7C) [57]. At the same time, the leaves of OE plants suffered less damage than VC plants after salt stress, and the results of Fv/Fm images and Fv/Fm values were consistent with this phenotype (Figure 7D,E). Therefore, OE plants were considered to be more salt-tolerant than VC plants.

### 2.8. The Effects of AvNAC030 Overexpression on ROS Scavenging in Arabidopsis 

ROS can reflect the degree of salt damage to plants, usually in the form of H_2_O_2_ and O^2−^, which can be directly reflected by the color after 3,3′-diaminobenzidine (DAB) and nitro blue tetrazolium (NBT) staining [58]. Therefore, in order to understand the ability of *AvNAC030* to scavenge ROS, the OE and VC *Arabidopsis* before and two days after treatment were histochemically stained with DAB and NBT. DAB and NBT staining revealed no significant difference between OE and VC *Arabidopsis* before treatment. After salt treatment, OE plants showed the lowest levels of brown precipitate and blue spots compared with VC plants (Figure 8A,B). Subsequently, the content of H_2_O_2_ and O^2−^ was detected, and the results were consistent with the results of the dyeing tests (Figure 8D,E). We then observed the cell death of OE and VC plants through trypan blue staining, and the cell death was related to the degree of damage caused by ROS (Figure 8C). These results indicate that overexpression of *AvNAC030* can effectively eliminate ROS and reduce the damage to plants caused by salt stress.

### 2.9. The Physiological Effects of AvNAC030 Overexpression in Arabidopsis

In order to study the function of *AvNAC030* after salt stress, we used OE and VC plants before and two days after salt treatment as materials to detect the indexes related to the ability to scavenge ROS and regulate osmoregulation substances. The results showed that the electrolyte leakage (EL) and malondialdehyde (MDA) values of OE plants were significantly lower than that those of VC plants after two days of salt stress, indicating that the cell membrane integrity was better preserved by OE plants following salt stress (Figure 9A,B). We then tested the multifunctional osmolytes and found that the proline content of OE plants was significantly higher than that of VC plants after salt treatment (Figure 9D). Similarly, the activity of SOD (superoxide dismutase), POD (peroxidase), and CAT (catalase) in OE plants was significantly higher than that in VC plants after salt stress (Figure 9D–E). These results indicated that overexpression of *AvNAC030* could effectively improve the salt tolerance of plants.

### 2.10. The Expression Analysis of Genes Involved in Salt Tolerance 

To further investigate the molecular mechanism of *AvNAC030* after salt stress, we measured the relative expression levels of marker genes related to salt stress. The results showed that after salt treatment, the expression levels of *AtMYB111*, *AtOZF1* (*Oxidation-related Zinc Finger 1*), *AtGSTU5* (*Glutathione S-transferase class tau 5*), and *AtP5CS1* (*delta1-pyrroline-5-carboxylate synthase 1*) in OE plants were significantly higher than those in VC and WT (Wild type) plants. These results suggest that *AvNAC030* may increase the salt tolerance of plants by regulating these salt stress-related genes (Figure 10).

## 3. Discussion

Kiwifruit has the effects of promoting digestion, lowering cholesterol, lowering blood lipids, enhancing immunity, preventing cancer, and being anticancer. It is known as the king of fruits and the king of vitamin C [59]. Although it is an emerging fruit tree, it has been developed rapidly in recent years. However, there are some restrictions in the process of industrial development. Kiwifruit is suitable for neutral acid soil, but soil salinization is increasing in kiwifruit production areas. It has adverse effects on the growth and development of plants, leading to the decline in yield and quality. Therefore, it is urgent to study its salt tolerance response mechanism and adaptation strategy, so as to provide a theoretical basis for the breeding of new kiwifruit varieties and the cultivation of resistant materials. In the preliminary study, the *A. valvata* germplasm material ZMH with strong salt tolerance was selected [34]. The root system of the material is well developed and has good compatibility as a rootstock for grafting the *valvata* Dunn, *A. chinensis* Planchon, *A. deliciosa* (Chev.) C. F. Liang & A. R. Ferguson, *A. arguta* (Siebold & Zucc.) Planch. ex Miq. Therefore, ZMH is a promising resource of resistant rootstocks, as well as a high-quality material for mining salt tolerance genes and studying the mechanism of salt tolerance regulation.

Taking ZMH as the research material, after removing the pseudogenes, we finally obtained 120 NAC family members. Then, we conducted a phylogenetic analysis and divided them into 13 subfamilies. The results were similar to those of *Arabidopsis* [44]. It has been reported that *NAC* genes in the same subgroup may have similar functions, such as specific resistance to stresses or plant specificity [60]. Liu et al. found that *ATAF1* in the *Arabidopsis* ATAF subfamily significantly improved the salt tolerance of transgenic rice [61]. Al-Abdallat et al. improved the salt tolerance of tomato by overexpressing two *ATNAC3*-related genes [62]. In addition to ATAF, the ATNAC3 subfamily and SENU5 subfamily have also been reported to respond to salt stress or improve plant salt tolerance [63]. *HaNAC-1* in the SENU5 subfamily from sunflower was observed to be upregulated in seedling roots and shoots in response to salinity stress [64]. *CarNAC1* from the SENU5 subfamily was strongly induced by salt stress [65]. *BnNAC5* from the SENU5 subfamily of *Brassica napus* is involved in response to high-salinity stress [66]. Dong et al. found that overexpression of *ClNAC9* in the SENU5 subfamily increased the saline resistance of transgenic *Arabidopsis* [67,68]. Liu et al. found that the *Chrysanthemum lavandulifolium* (Fisch. Ex Trautv.) Makino gene *ClNAC9* in the SENU5 subfamily positively regulated saline stress in transgenic *chrysanthemum grandiflora* Hook [69]. Wang et al. found that overexpressing the NAC transcription factor *LpNAC13* of the SENU5 subfamily from *Lilium pumilum Redouté* in tobacco positively regulated the salt response [70]. According to the phylogenetic relationship, *AvNAC030* belongs to the SENU5 subfamily (Figure 1). The results of motif analysis provide further evidence for this phylogenetic relationship (Figure 2). Interestingly, most of the conserved motifs are at the N-terminus of the NAC domain, which is consistent with the previous description, indicating that these motifs are necessary for the function of NAC [71]. The results of conserved amino acid residues show that the C and D subdomains are relatively conserved, indicating that the NAC family has retained its basic functions during long-term evolution. The variability of the B and E subdomains illustrates their importance in functional diversity (Figure 3). The analyses of the expression pattern and interaction network show that *AvNAC030* responds to salt stress (Figure 4 and Figure 5). These results suggest that *AvNAC030* plays a key role in the regulation mechanism of salt tolerance.

The result of subcellular localization show that *AvNAC030* may function as a transcription factor (Figure 6). To understand the regulatory mechanism of *AvNAC030*, we also used transgenic *Arabidopsis* to study its function after salt stress. We found that after salt treatment, OE significantly reduced the damage caused by salt stress compared with VC plants (Figure 7D,E). Therefore, the survival rate of OE plants was higher than that of VC plants (Figure 7A,B). ROS usually exists in the form of H_2_O_2_ and O^2−^, and can be rapidly produced in plants when exposed to adverse environmental conditions such as high salinity, drought, or extreme temperatures [72]. Excessive ROS leads to oxidative damage of cell components such as proteins, lipids, and DNA. Plants maintain the balance between ROS production and removal to ensure ROS homeostasis, thereby reducing the effects of oxidative stress [73]. Flavonoids, as non-enzymatic antioxidants, have been widely reported to reduce ROS damage in plant cells under biotic and abiotic stress [74]. After being exposed to salt stress, OE plants accumulated more flavonoids than VC plants. We then tested their ability to eliminate H_2_O_2_ and O^2−^. The ROS scavenging ability of OE plants was superior to that of VC plants, and more living cells were retained (Figure 8). The results were consistent with the phenotype. MDA, as a decomposition product of polyunsaturated fatty acids, has a positive correlation with the accumulation of ROS [75]. The results showed that after being exposed to salt stress, the cell membrane of VC plants was damaged by salt to a higher degree, resulting in more soluble leakage, and therefore had a higher EL value and MDA content (Figure 9A,B). Proline plays an important role in scavenging hydroxyl radicals. In addition, it stabilizes the subcellular structure and protects cellular macromolecules against damage by adjusting the intracellular osmotic potential [76]. SOD can catalyze the conversion of superoxide anions into H_2_O_2_ and O_2_, and is an important material for scavenging free radicals in plants, while POD and CAT are enzymes for scavenging H_2_O_2_. SOD, POD, and CAT maintain the steady level of free radical content in plants through synergistic action, and prevent the changes in plant physiology and biochemistry caused by free radicals [77]. The results of determining proline content and SOD, POD, and CAT activities showed that OE plants had a stronger ability to scavenge ROS than VC plants under salt stress (Figure 9C–F). *AtMYB111* improves ROS scavenging efficiency by regulating the synthesis of flavonoids [25]. *AtOZF1* plays a role in regulating oxidative stress response in *Arabidopsis* [78]. *AtGSTU5* is used as a marker of oxidative stress [79]. *AtP5CS1* is a proline synthesis marker gene [80]. The results show that *AvNAC030* might enhance the salt tolerance of plants by regulating these stress-related genes after salt stress (Figure 10). These results suggest that *AvNAC030* can increase the salt tolerance of plants by improving the efficiency of ROS removal and maintaining the intracellular and extracellular osmotic balance to protect the integrity of the membrane (Figure 11).

## 4. Materials and Methods

### 4.1. Sequence Retrieval and Identification of A. valvata NAC Genes

The NAC sequences of *Arabidopsis* were obtained from TAIR (https://www.arabidopsis.org/, accessed on 6 May 2020). The NAC sequences of kiwifruit were retrieved from the full-length transcriptomic data of ZMH (unpublished). We removed repetitive sequences and incomplete sequences. The retrieved NACs were screened by analyzing the conserved domain using the conserved domains database (https://www.ncbi.nlm.nih.gov/Structure/cdd/wrpsb.cgi, accessed on 8 June 2020). The obtained sequences containing the conserved NAC domain (PF02365) were detected again by the Pfam database (http://pfam.xfam.org/, accessed on 8 June 2020). The details of the NAC family were obtained by the ExPASy Proteomics server (http://web.expasy.org/compute_pi/, accessed on 12 June 2020) (Table 3). Nucleotide and amino acid sequences based on the full-length transcriptome data are presented in Table A1 (Appendix A).

### 4.2. Bioinformatic Analysis of the NAC Family in Kiwifruit

We used Clustal Omega (http://www.ebi.ac.uk/Tools/msa/clustalo/, accessed on 10 July 2020) for phylogenetic analysis, which was then presented with the Interactive Tree of Life (iTOL) (https://itol.embl.de/itol.cgi, accessed on 12 July 2020). The numbers were bootstrap values based on 1000 iterations. Only bootstrap values larger than 50% support were displayed. We identified the conserved motifs with MEME (http://meme-suite.org/index.html, accessed on 16 July 2020) and retained e-values < 1 × 10^−20^ for analysis. We performed multiple sequence alignments of AvNACs using CLUSTALW (https://myhits.sib.swiss/cgi-bin/clustalw, accessed on 18 July 2020) with default parameters. The heatmap of the NAC family was generated with TBtools (https://github.com/CJ-Chen/TBtools/releases, accessed on 21 July 2020) based on the ZMH RNA-seq data (unpublished). The prediction of the gene interaction network was completed by STRING (https://string-db.org/cgi/input.pl, accessed on 26 July 2020) with option value > 0.700.

### 4.3. The Sample Collection

The ZMH from *A. valvata* was grown in a greenhouse at the Zhengzhou Fruit Research Institute, Chinese Academy of Agricultural Sciences, Zhengzhou, Henan Province, China (34°43′ N, 113°39′ E, altitude 111 m). When the height of the tissue culture seedlings reached 40 cm, they were treated with 0.4% NaCl solution. Samples were taken for sequencing after treatment at 0 (I), 12 (II), 24 (III), and 72 (IV) h. Each sample had three biological replicates and each replicate included roots from three plants.

### 4.4. Subcellular Localization

The open reading frame (ORF) of *AvNAC030* with a mutational stop codon was cloned between the Xba I and Sal I sites of the pB221-GFP vector with the T4 DNA ligase (Thermo Scientific, Waltham, MA, USA) and a pair of primers (Table 4). Protoplasts were prepared from rosette leaves of 4-week-old *A.*
*Arabidopsis* seedlings, and the recombinant and control plasmids were transformed into *Arabidopsis* protoplasts by using PEG (polyethylene glycol) 4000 mediated transformation [81]. The N-terminal of *AtBZR2* (*AT1G19350.3*) contained an NLS, so we fused it with mCherry to label the nuclear of protoplast. [82]. After 18 h, the GFP fluorescence was observed under a laser scanning confocal microscope (Olympus FV1000 viewer, Tokyo, Japan).

### 4.5. The Transformation of Arabidopsis and Stress Treatments

The RNA was extracted using TIANGEN RNAprep Pure Plant Kit (Tiangen Biotech, Beijing, China). We used DNase I (Thermo Scientific, Waltham, MA, USA) to remove genomic DNA from total RNA and a RevertAid First Strand cDNA Synthesis Kit (Thermo Scientific, Waltham, MA, USA) to synthesize the first-strand complementary DNA (cDNA) according to the manufacturer’s instructions. The full-length ORF of *AvNAC030* was obtained by the primers (Table 4) and Pfu DNA polymerase (TransGen Biotech, Beijing, China). The products were purified and integrated into the blunt vector (pEASY-Blunt Simple Cloning Kit, Beijing, China) for sequencing, and then was cloned between the Nco I and Bgl II sites of the pCAMBIA3301 vector with the T4 DNA ligase and a pair of primers (Table 4). The floral dip method was used for genetic transformation, and phosphinothricin resistance was used to detect positive plants. The homozygous T_3_ generation was germinated in soil chambers in a greenhouse at 22 °C with 16 h light/8 h dark cycle and 70% relative humidity, and the four-week-old potted *Arabidopsis* plants were subjected to 250 mM NaCl treatment for functional verification.

### 4.6. Histochemical and Physiological Analysis

Total flavonoid content was measured as described previously by Jia [83]. For chlorophyll fluorescence measurements, the images were obtained by IMAGING-PAM chlorophyll fluorometer (Walz, Effeltrich, Germany), and the maximum quantum efficiency of photosystem II (Fv/Fm) was measured with Imaging WinGegE software [84]. H_2_O_2_ and O_2_^−^ were stained with DAB (Solarbio, Beijing, China) and NBT (Beijing Biodee Biotechnology, Beijing, China). The programmed cell death was detected by 0.4% trypan blue solution (MYM Biological Technology Company Limited, Chicago, IL, USA). The H_2_O_2_ and O_2_^−^ content was measured as described previously by Liu and Elstner [85,86]. EL was measured as described previously by Ben-Amor [87]. MDA, proline, SOD, POD, and CAT activity were detected using corresponding test kits (Nanjing Jiancheng Bioengineering Institute, Nanjing, China) [88].

### 4.7. RT-qPCR Analysis

qRT-PCR was performed in the presence of SYBR green qPCR Master Mix (Fermentas, Ontario, Lithuania) and the amplification was performed in the Eco Real-Time PCR system (Illumina, San Diego, CA, USA). All reactions were performed in triplicate. The primers were designed using Oligo 7.0 and are listed in Table 4.

### 4.8. Statistical Analysis

All experiments were replicated independently at least three times, and data are shown as the mean ± SD of three independent experiments. Data were subjected to analysis of variance (ANOVA) using the Statistical Analysis System (SPSS version 22.0) software. The differences between the means were compared using the Tukey’s test (*p* < 0.05).

## 5. Conclusions

Using ZMH as the material, we performed high-throughput sequencing at the four time points after its salt treatment. We then analyzed the members of the NAC family based on the sequencing results and bioinformatics analysis. According to the results, we speculate that *AvNAC030* may play a positive role in the mechanism of salt tolerance. Finally, we used *Arabidopsis* genetic transformation technology and combined it with phenotype, physiology and molecular biology to analyze the function of *AvNAC030* under salt stress. In this way, we can fully explore the original data and combine bioinformatics analysis with molecular biology experiments more efficiently to study the function of the *NAC* family.

## Figures and Tables

**Figure 1 ijms-22-11897-f001:**
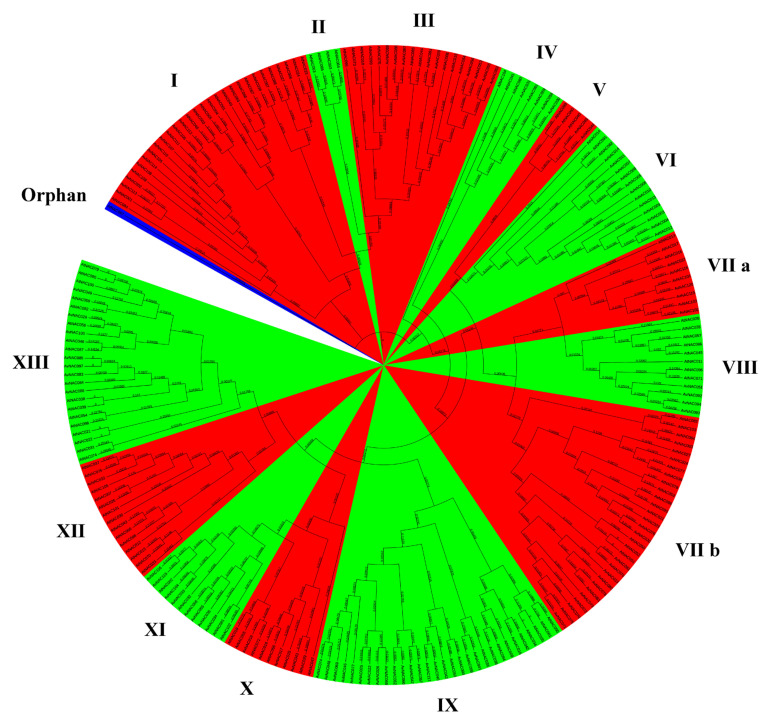
Phylogenetic analysis of the NAC transcription factors of kiwifruit and *Arabidopsis*.

**Figure 2 ijms-22-11897-f002:**
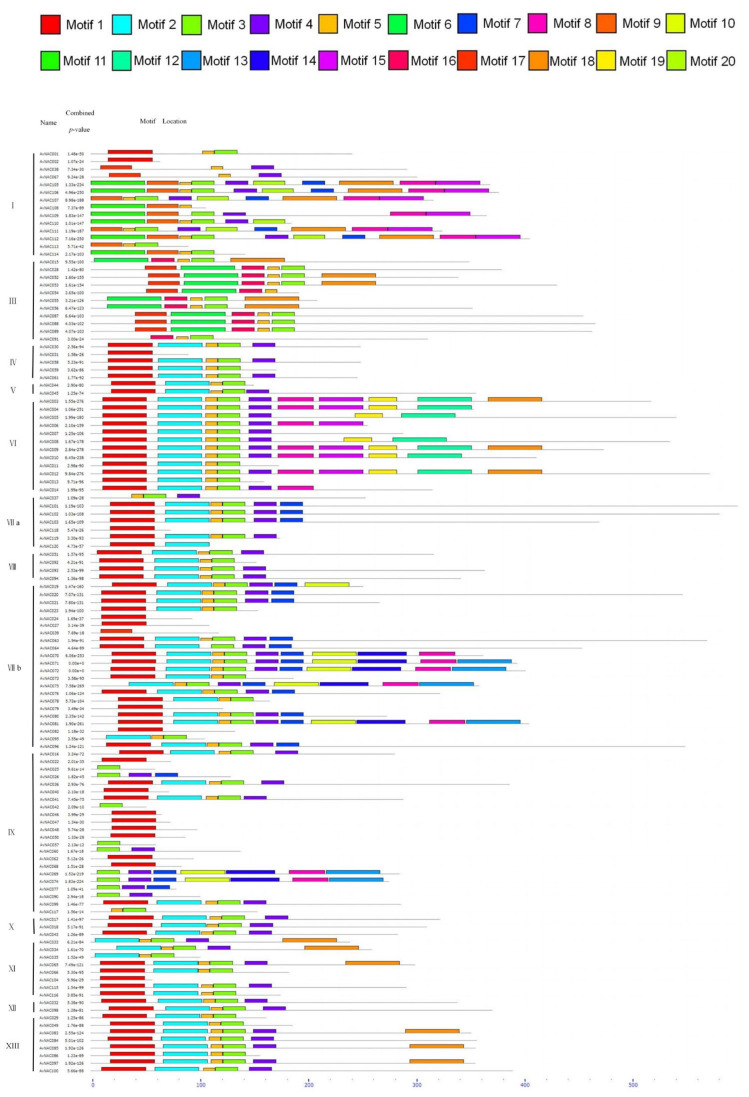
The conservative motif analysis of the NAC family in kiwifruit. The rectangles of different colors represent different conservative motifs, and black lines represent the non-conserved sequences.

**Figure 3 ijms-22-11897-f003:**
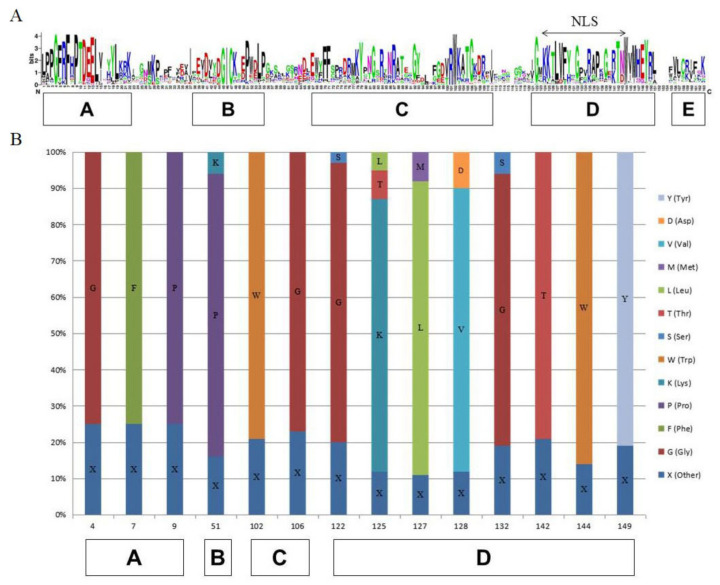
The sequence analysis of five subdomains in kiwifruit. (**A**) Sequence logo of five subdomains in kiwifruit. (**B**) The conserved amino acid distribution in five subdomains in kiwifruit. The histogram shows the percentage of amino acids at this position.

**Figure 4 ijms-22-11897-f004:**
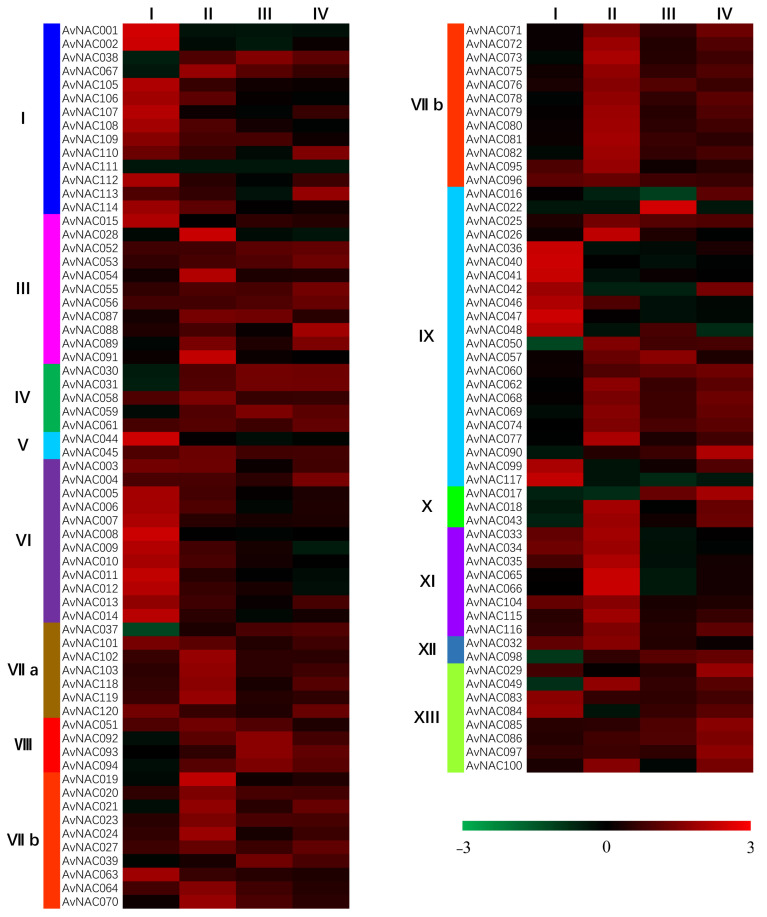
The heatmap of the NAC family at different time points after salt treatment.

**Figure 5 ijms-22-11897-f005:**
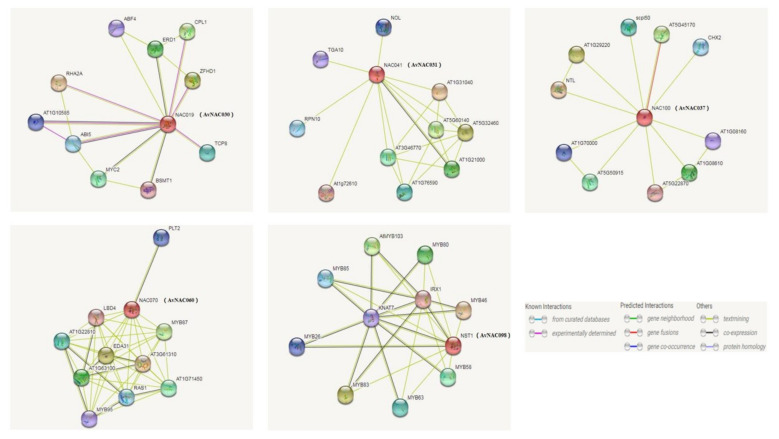
The interaction network analysis for *AvNAC030*, *AvNAC031*, *AvNAC037*, *AvNAC060*, and *AvNAC098*. Note: *AvNAC* genes are shown in brackets.

**Figure 6 ijms-22-11897-f006:**
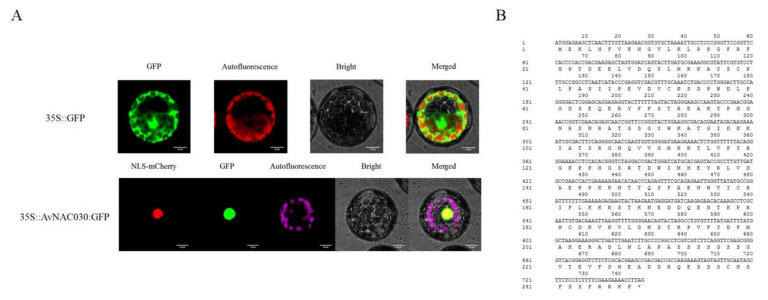
Subcellular localization of *AvNAC030.* (**A**) The vector control (*35S::GFP*) and fusion protein construct *35S::AvNAC030:GFP* were introduced into the *Arabidopsis* protoplast. (**B**) CDS and peptide sequence of *VvSAUR041*.

**Figure 7 ijms-22-11897-f007:**
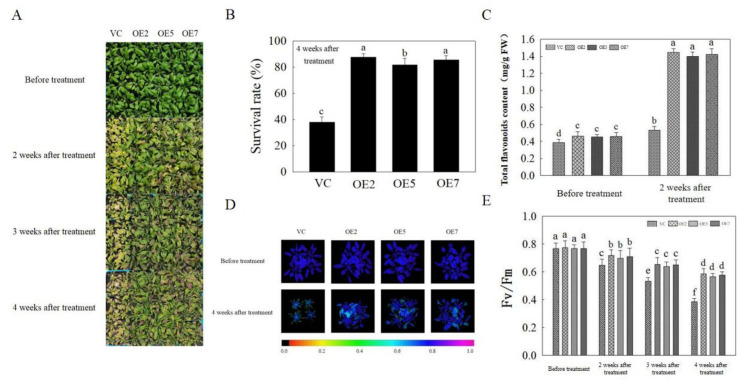
Phenotypic and physiological indexes of vector control (VC) and overexpression (OE) *Arabidopsis* under normal and stress conditions. (**A**) The phenotypic of VC and OE *Arabidopsis* under normal and stress conditions. (**B**) The survival rate of VC and OE *Arabidopsis* after salt stress. (**C**) The total flavonoid contents of VC and OE *Arabidopsis* under normal and stress conditions. (**D**) The Fv/Fm images of VC and OE *Arabidopsis* under normal and stress conditions. (**E**) The Fv/Fm value of VC and OE *Arabidopsis* under normal and stress conditions. Different letters represent significant difgerences (*p* < 0.05).

**Figure 8 ijms-22-11897-f008:**
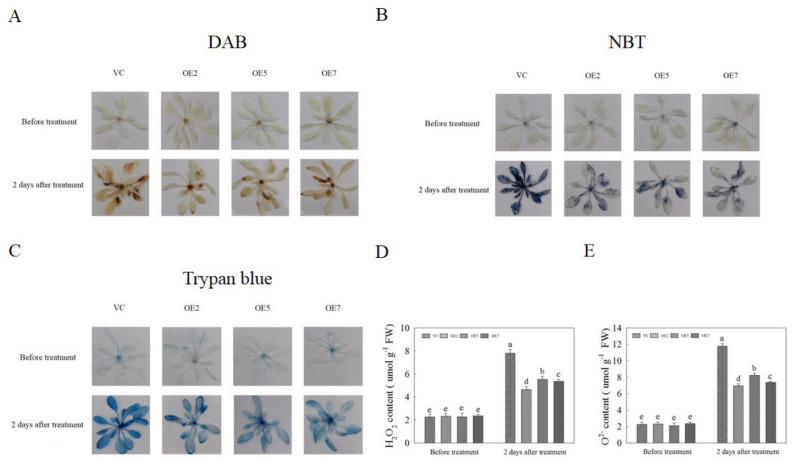
Reactive oxygen species (ROS) scavenging ability and cell death of vector control (VC) and overexpression (OE) *Arabidopsis* under normal and stress conditions. (**A**) 3,3′-diaminobenzidine (DAB) staining. (**B**) Nitro blue tetrazolium (NBT) staining. (**C**) Trypan blue staining. (**D**) H_2_O_2_ content. (**E**) O^2−^ content. Different letters represent significant difgerences (*p* < 0.05).

**Figure 9 ijms-22-11897-f009:**
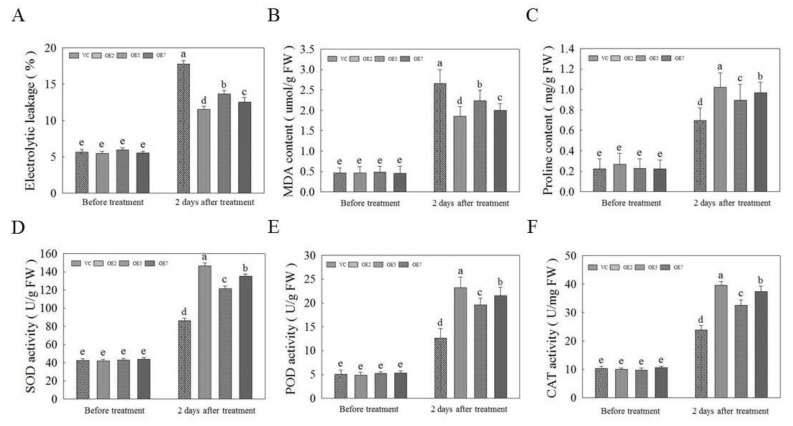
Antioxidant and osmotic indices of vector control (VC) and overexpression (OE) *Arabidopsis* under normal and stress conditions. (**A**) Electrolytic leakage. (**B**) Malondialdehyde (MDA) content. (**C**) Proline content. (**D**) Superoxide dismutase (SOD) activity. (**E**) Peroxidase (POD) activity. (**F**) Catalase (CAT) activity. Different letters represent significant difgerences (*p* < 0.05).

**Figure 10 ijms-22-11897-f010:**
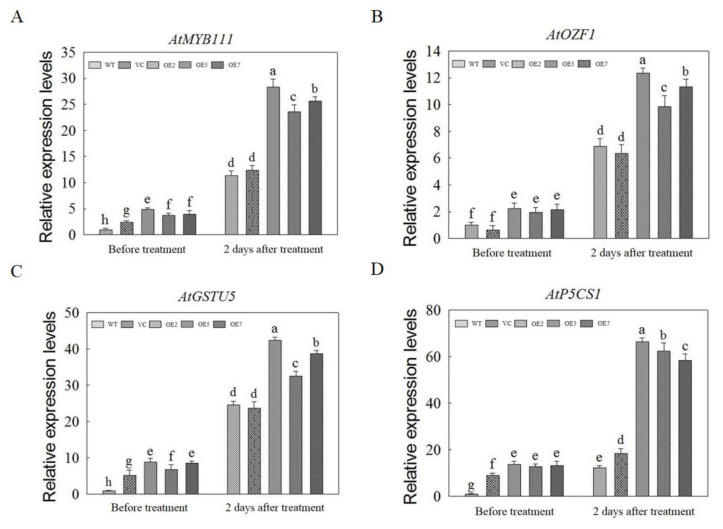
The relative expression levels of salt stress-related genes. (**A**) The relative expression levels of *AtMYB111*. (**B**) The relative expression levels of *AtOZF1.* (**C**) The relative expression levels of *AtGSTU5*. (**D**) The relative expression levels of *AtP5CS1*. Different letters represent significant difgerences (*p* < 0.05).

**Figure 11 ijms-22-11897-f011:**
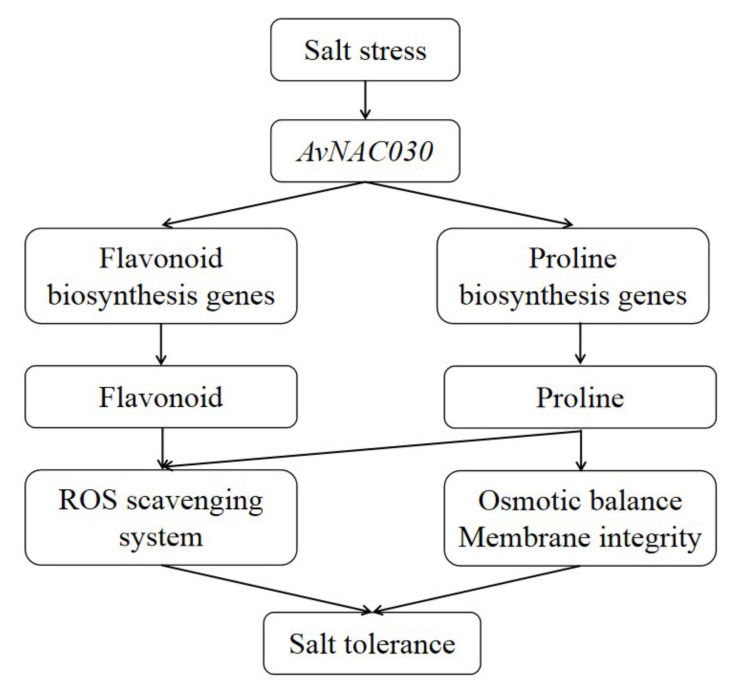
The hypothesis of the regulatory network of the AvNAC030 involved in salt stress responses.

**Table 1 ijms-22-11897-t001:** Subfamily classification adjustment.

Family Number	Family Name
I	ANAC001
II	ANAC063
III	ONAC003
IV	SENU5
V	OsNAC8
VI	TIP
VII a	NAC2
VIII	ANAC011
VII b	NAC2
IX	TERN, ONAC022
X	NAP, AtNAC3
XI	ATAF
XII	OsNAC7
XIII	NAC1, NAM

**Table 2 ijms-22-11897-t002:** Normal expression sequences of 20 motifs identified. Sites represent the time of motif appeared.

Motif	Sequences	Sites	Length (aa)
Motif1	G[FY][RK]F[HRSL]PT[DE][EKQ][EQ]L[VIL][QVGDIN][YHQ]YL[KRCM][RSN][KR][IAV][CSNY][GSD][KDL][PER][FLI][RAP][VFL] [ED][VI][IV][SARP][EDV][VIT][DE][VLI][YCN][KH][SQF][ED]P[WLE]	80	41
Motif2	[GRN][DE][LKDR][EQ][WYR][YF]FF[SC][PLT][RKVL][DE][RKA]KY[PGQ][NT]G[SQVA]R[TLPS]NR[AV][TA][EGRK][RSTK]G[YF] WK[AT]TG[KAR]D[RK][TPKS][IV][RFV][SHC]	68	41
Motif3	[FYV][YH][RKSGENA][GV][RK][AG]P[KR]G[EKQTI][RK]T[ND]W[VI][MI][HQ][EQ]Y[RHT][LIA]	98	21
Motif4	[GKQSP][PKVTS][AGST][QLKA][DEG][ASD][FWYR][VAL][LVI][CY][RK][IVL][FIY][QKRL][KS][SKHN][GDERA][SLE][GKSIA] [PKVE][KNPE]	68	21
Motif5	[SGTPN][KQARVE][VTLA][IVTK]G[MCVI][KR][KT][TIA][LM][VD]	91	11
Motif6	H[PM][FL]IDEFIPT[VI][DKG][EGR][DE][DE]GICYTHP[QE][NKY]LPG[VA][KTR][QRT]DG[SLN][VS][SKYV]HFFH[RI][API][IS] [KNM]AY[NTA]TG[TQ]R	10	50
Motif7	[DQP][YRG][GY][AL][PQ][FI][IRKV][EPD]E[ED][WT][DNEA][DNE][DE][DVEA][SLCPNE][IVMLTF][VSMILCY][VIRDG][PGNQ][GDS]	25	21
Motif8	[VMTN][ER][IPAFH][SPA][NLK][LC][EGD][SD][LV][EVD][PKQ][KTN][EDFV][NSG][RSH][PK][NSEGA][KPTNI][VLA][KTE][AS] [IF][DTS][ESKDA][DNAT][FMNI][LQ][ESD][KE][PKSF]V[PT][PE]	20	33
Motif9	[MS]WYL[LI]RS[DE][HN][KN]KNSEHGFW[TR]ARG[DE][AG][SI]EIFM	10	29
Motif10	VN[AG]DD[SA][QK]VEGN[DE][FYL]EQD[TI]HS[HT]N[MK][AS][AP]L[YRC][QL][TA]EL[PQ][NI][GVLF][CSL][QNE][NTL][IVF] [PH][FL][FV][CFA]	8	41
Motif11	MAPRPRDSIGLYW[AT]D[EA][EA]IIMSLE[RGE]MEKGSP[IN]P[VE]NVSVDVNPYQ[YC]KP[LI][NY]L[PR]	7	50
Motif12	[QN][FD][QP][NY]GTNES[GVI]S[YSE]QN[ML][AS]VEI[EV]E[LYP]NYLN[IT][MV][NDS][FNI][LF]DKE[TI]GSCS[ED]SDADV [AT]QAQ[IVF]	7	50
Motif13	YLKFI[NS]NLENEILNVSMERETLKIE[VL]MRAQAMIN[VMI]LQ[SL]RI[DE][LV][LV]N[KR][ED]NE[DE][LR]	6	50
Motif14	RE[TR][SP][GE][YD][AC]P[FL]P[CG][TI][VA][DN][AP]E[PT][IL]S[VL]VPNK[KR][ST][RK][HN][DE][DN]PNSSNANGSEDS[TN]TT	7	45
Motif15	[AN][EA][KI][RLQ]S[KT][TAK][SMA][DR][SNK][CWR][LN][EA][EG]TS[DNI][ST][TIH][FND][IVA]A[TD][DS][AS][GRS][YSG][EH] [KS][AT][IF]P[PRV][DGRE][EYRK][KEA][KLN][ESFA][VET][AMY][GLN]	12	41
Motif16	[KTN][IEQ][HCQD][GDS][DLM][DQTK][FGK][GEC][DEG][VT]RWHKTG[RK]T[KR][PC]V	11	21
Motif17	[EMD][WA][PVFL][GMS][LV]P[AYRT]G[VF][KRT]F[DNS]P[STKE]D[QVH]E[LI][IML][EYW][HLI]L[LEYA][AR]K[VAC][EGNRH] [AVGI][GNK]	10	29
Motif18	[SVFYM][TSYNF][QSWTY][QSE][DQRE][NQIHE][SVGH][KSQNVA][SQCE][TVSED][DTSPH][KPSI][FTWR][ENAS][DHSNG] [EGDA][LF][PFGEA][DNQ][SGNEA][LTNFA][DAVH][LVYF][LMIQG][IRTQE][ADPN][LNYEA][VGML][ADT][APKV][VISNF] [STN][QPNAE][DQSA][NSEP][KVQHF][TGVSA][PNYAQ][DQNE][PATLM][NPF][RVQN][KTFSQ][ELPMD][SKTEM][QVME] [LAIDF][HASCM][CSFPL][ILSF]	19	50
Motif19	PN[QL]QAP[DY][CG][ND]GKIFSPVH[VRK]QMQ[TM]EL[GA][SY]	7	26
Motif20	NVG[DG]CTGSNDIHPSVVPKSG[SN]TSGQGCMS	6	29

**Table 3 ijms-22-11897-t003:** Details of the *VvSAUR* family. Mw, molecular weight; PI, isoelectric point.

Gene ID	Gene Symbol	ORF Length (bp)	No. of aa	Mw	PI	Group
R_transcript_18612	*AvNAC001*	723	240	27,212.06	9.75	I
R_transcript_34811	*AvNAC002*	192	63	7188.37	6.9	I
R_transcript_33699	*AvNAC003*	1554	517	58,292.66	5.05	VI
R_transcript_37169	*AvNAC004*	1062	353	40,182.48	4.75	VI
R_transcript_40690	*AvNAC005*	1623	540	60,232.03	4.8	VI
R_transcript_52636	*AvNAC006*	768	255	29,280.8	5.48	VI
R_transcript_59072	*AvNAC007*	867	288	33,147.17	8.47	VI
R_transcript_63416	*AvNAC008*	1605	534	59,209.84	4.7	VI
R_transcript_66645	*AvNAC009*	1422	473	53,702.72	5.07	VI
R_transcript_69568	*AvNAC010*	1236	411	46,871.81	4.9	VI
R_transcript_71270	*AvNAC011*	522	173	20,016.13	9.97	VI
R_transcript_79093	*AvNAC012*	1716	571	64,058.05	4.93	VI
R_transcript_8696	*AvNAC013*	480	159	18,788.47	9.66	VI
R_transcript_9544	*AvNAC014*	948	315	35,366.8	5.25	VI
R_transcript_99187	*AvNAC015*	1047	348	39,632.89	4.75	III
R_transcript_46831	*AvNAC016*	840	279	32,147.09	5.65	IX
R_transcript_25400	*AvNAC017*	966	321	35,794.65	8.74	X
R_transcript_18707	*AvNAC018*	930	309	34,897.28	7.73	X
R_transcript_44451	*AvNAC019*	756	251	28,292.87	7.13	VII b
R_transcript_13572	*AvNAC020*	1641	546	60,679.5	4.67	VII b
R_transcript_33721	*AvNAC021*	801	266	30,395.54	5.72	VII b
R_transcript_44030	*AvNAC022*	222	73	8480.77	6.56	IX
R_transcript_62201	*AvNAC023*	465	154	18,056.56	9.45	VII b
R_transcript_66399	*AvNAC024*	282	93	10,565.43	8.01	VII b
R_transcript_81784	*AvNAC025*	177	58	6986.9	8.11	IX
R_transcript_86796	*AvNAC026*	387	128	14,679.38	4.72	IX
R_transcript_92715	*AvNAC027*	327	108	12,554.41	7.92	VII b
R_transcript_86654	*AvNAC028*	1137	378	42,536.75	5.67	III
R_transcript_80114	*AvNAC029*	846	161	18,976.48	9.35	XIII
R_transcript_71454	*AvNAC030*	747	248	28,221.73	9.18	IV
R_transcript_78474	*AvNAC031*	270	89	10,261.05	7.7	IV
R_transcript_35688	*AvNAC032*	1014	337	39,025.5	5.05	XII
R_transcript_40217	*AvNAC033*	717	238	27,446.92	7.07	XI
R_transcript_50816	*AvNAC034*	777	258	29,314.12	8.43	XI
R_transcript_52293	*AvNAC035*	303	100	11,324.11	10.76	XI
R_transcript_83913	*AvNAC036*	1158	385	43,238.02	6.12	IX
R_transcript_27414	*AvNAC037*	762	253	28,895.91	8.72	VII a
R_transcript_38643	*AvNAC038*	876	291	33,401.71	6.56	I
R_transcript_67877	*AvNAC039*	354	117	13,809.75	9.51	VII b
R_transcript_65045	*AvNAC040*	216	71	8213.49	4.86	IX
R_transcript_75291	*AvNAC041*	864	287	33,087.95	6.47	IX
R_transcript_94887	*AvNAC042*	153	50	5808.92	9.69	IX
R_transcript_61978	*AvNAC043*	849	282	32,403.68	8.7	X
R_transcript_17613	*AvNAC044*	453	150	17,230.56	8.98	V
R_transcript_19469	*AvNAC045*	1068	355	40,547.75	8.04	V
R_transcript_22363	*AvNAC046*	195	64	7366.41	4.47	IX
R_transcript_30245	*AvNAC047*	219	72	8494.58	4.64	IX
R_transcript_50769	*AvNAC048*	294	97	11,033.81	4.78	IX
R_transcript_78204	*AvNAC049*	558	185	21,592.1	9.25	XIII
R_transcript_98040	*AvNAC050*	261	86	9774.18	4.9	IX
R_transcript_38748	*AvNAC051*	951	316	36,191.6	8.06	VIII
R_transcript_28167	*AvNAC052*	1017	338	37,889.53	5.96	III
R_transcript_31813	*AvNAC053*	1290	429	48,120.63	4.89	III
R_transcript_40244	*AvNAC054*	576	191	21,494.66	9.18	III
R_transcript_62357	*AvNAC055*	627	208	23,465.75	7.73	III
R_transcript_99111	*AvNAC056*	1056	351	39,828.16	4.84	III
R_transcript_18002	*AvNAC057*	180	59	6981.22	10.21	IX
R_transcript_19545	*AvNAC058*	747	248	28,318.85	8.97	IV
R_transcript_46194	*AvNAC059*	513	170	19,518.45	9.56	IV
R_transcript_95592	*AvNAC060*	414	137	15,709.54	8.81	IX
R_transcript_53223	*AvNAC061*	738	245	27,771.36	9.62	IV
R_transcript_54724	*AvNAC062*	285	94	10,901.71	9.17	IX
R_transcript_56133	*AvNAC063*	1704	567	62,826.9	5.04	VII b
R_transcript_90698	*AvNAC064*	1359	452	50,328.74	4.7	VII b
R_transcript_19894	*AvNAC065*	897	298	34,337.97	6.26	XI
R_transcript_56139	*AvNAC066*	549	182	20,833.91	9.94	XI
R_transcript_27385	*AvNAC067*	903	300	33,847.25	6.33	I
R_transcript_9009	*AvNAC068*	252	83	9368.84	5.57	IX
R_transcript_9620	*AvNAC069*	855	284	32,050.91	4.94	IX
R_transcript_16576	*AvNAC070*	1086	361	40,769.42	5.32	VII b
R_transcript_16797	*AvNAC071*	1179	392	44,361.79	5.5	VII b
R_transcript_16893	*AvNAC072*	1203	400	45,569.26	5.72	VII b
R_transcript_37237	*AvNAC073*	561	186	21,404.4	9.64	VII b
R_transcript_56738	*AvNAC074*	825	274	30,781.31	4.7	IX
R_transcript_69341	*AvNAC075*	1074	357	40,361.19	5.65	VII b
R_transcript_13655	*AvNAC076*	966	321	36,708.61	4.62	VII b
R_transcript_15641	*AvNAC077*	237	78	8818.97	7.93	IX
R_transcript_80139	*AvNAC078*	495	164	18,695.36	9.69	VII b
R_transcript_82604	*AvNAC079*	366	121	13,582.7	5.58	VII b
R_transcript_94099	*AvNAC080*	819	272	30,604.57	6.79	VII b
R_transcript_95060	*AvNAC081*	1212	403	45,245.69	5.46	VII b
R_transcript_100635	*AvNAC082*	399	132	14,534.98	4.95	VII b
R_transcript_54585	*AvNAC083*	1053	350	38,925.86	8.52	XIII
R_transcript_86053	*AvNAC084*	1068	355	39,869.89	8.72	XIII
R_transcript_90949	*AvNAC085*	1065	354	39,261.19	8.52	XIII
R_transcript_96411	*AvNAC086*	468	155	17,988.47	9.44	XIII
R_transcript_42641	*AvNAC087*	1362	453	50,691.58	6.44	III
R_transcript_50235	*AvNAC088*	1395	464	51,813.64	6.45	III
R_transcript_94297	*AvNAC089*	1386	461	51,455.33	6.49	III
R_transcript_73092	*AvNAC090*	303	100	11,338.9	7.87	IX
R_transcript_92394	*AvNAC091*	933	310	34,580.16	5.19	III
R_transcript_12933	*AvNAC092*	459	152	17,906.55	9.26	VIII
R_transcript_63861	*AvNAC093*	1092	363	40,740.4	4.93	VIII
R_transcript_85819	*AvNAC094*	1026	341	38,403.75	5.09	VIII
R_transcript_58057	*AvNAC095*	315	104	12,205.08	10.04	VII b
R_transcript_79749	*AvNAC096*	1644	547	61,366.06	4.59	VII b
R_transcript_14929	*AvNAC097*	1065	354	39,261.19	8.52	XIII
R_transcript_101459	*AvNAC098*	1110	369	42,049.87	6.14	XII
R_transcript_39496	*AvNAC099*	858	285	33,460.35	6.47	IX
R_transcript_95695	*AvNAC100*	1167	388	43,858.11	6.52	XIII
R_transcript_13398	*AvNAC101*	1794	597	66,986.19	4.94	VII a
R_transcript_95502	*AvNAC102*	1743	580	64,771.09	4.8	VII a
R_transcript_100431	*AvNAC103*	1410	469	53,408.4	4.68	VII a
R_transcript_68016	*AvNAC104*	171	56	6527.51	4.97	XI
R_transcript_15938	*AvNAC105*	1104	367	40,122.95	6.27	I
R_transcript_24316	*AvNAC106*	1128	375	41,234.35	5.86	I
R_transcript_30973	*AvNAC107*	948	315	34,509.58	6.54	I
R_transcript_31867	*AvNAC108*	318	105	12,183.73	5.06	I
R_transcript_41086	*AvNAC109*	1095	364	40,848.7	8.11	I
R_transcript_46223	*AvNAC110*	555	184	20,798.46	6.08	I
R_transcript_65760	*AvNAC111*	972	323	35,335.58	6.54	I
R_transcript_98265	*AvNAC112*	1215	404	44,680.32	5.57	I
R_transcript_98456	*AvNAC113*	270	89	10,625.08	9.06	I
R_transcript_100689	*AvNAC114*	426	141	16,454.73	6.29	I
R_transcript_19510	*AvNAC115*	873	290	32,994.47	7.65	XI
R_transcript_88888	*AvNAC116*	525	174	19,880.98	9.73	XI
R_transcript_27781	*AvNAC117*	462	153	17,111.21	8.93	IX
R_transcript_33085	*AvNAC118*	222	73	8541.94	5.28	VII a
R_transcript_34849	*AvNAC119*	525	174	20,351.13	9.31	VII a
R_transcript_47733	*AvNAC120*	330	109	12,696.51	9.51	VII a

**Table 4 ijms-22-11897-t004:** List of primers used for RT-qPCR and the construction of recombinant plasmids.

Gene Name	Gene Identifier	Forward Primer (5′-3′)	Reverse Primer (5′-3′)	The Purpose
*AvNAC030*	R_transcript_71454	ATGGAGAAGCTCAACTTTGTT	CTAAGGTTTTCTTCGAAAAGA	Obtain ORF
*AvNAC030-pB221-GFP*	R_transcript_71454	TCTAGAATGGAGAAGCTCAACTTTGTT	GTCGACCTAAGGTTTTCTTCGAAAAGA	Subcellular localization
*AvNAC030-3301*	R_transcript_71454	CCATGGATGGAGAAGCTCAACTTTGTT	AGATCTCTAAGGTTTTCTTCGAAAAGA	Expression vector construction
*AtMYB111*	At5g49330	GAACAAGGAAGCGAGACAAAG	TCCCAATCAAGCAACTCCTC	RT-qPCR
*AtOZF1*	At2g19810	TTCTGAAGATCTAACGGTGTC	CGGGATGAGCGTAAGGACACT	RT-qPCR
*AtGSTU5*	At2g29450	ATGGCTGAGAAAGAAGAAGTGAAGC	TTAAGAAGATCTCACTCTCTCTGCC	RT-qPCR
*AtP5CS1*	AT2G39800	TAGCACCCGAAGAGCCCCAT	TTTCAGTTCCAACGCCAGTAGA	RT-qPCR
*AtUBQ3*	AT5G03240	CGGAAAGACCATTACTCTGGA	CAAGTGTGCGACCATCCTCAA	RT-qPCR

## Data Availability

Sequence data from this work can be found in the NCBI database (SRA data).

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
