# Peer review of "AvNAC030, a NAC Domain Transcription Factor, Enhances Salt Stress Tolerance in Kiwifruit"

_ijms, 2021, doi:10.3390/ijms222111897_

Round 1

Reviewer 1 Report

In this manuscript, the author explains that the AvNAC030, a NAC domain transcription factor, enhances salt stress tolerance in kiwifruit. The authors have identified 120 NAC members and divided them into 13 subfamilies according to phylogenetic analysis. Subsequently, we conducted a comprehensive and systematic analysis based on the conserved motifs, key amino acid residues in the domain, expression patterns, and protein interaction network predictions and screened the candidate gene AvNAC030. In order to study its function, authors adopted the method of heterologous expression in Arabidopsis. Compared with the control, the overexpression plants had higher osmotic adjustment ability and improved antioxidant defense mechanism. These results suggest that AvNAC030 plays a positive role in the salt tolerance regulation mechanism in kiwifruit. The manuscript is very well written and has solid data. For the betterment of this manuscript, I have a few queries and suggestions for the authors. 1. The introduction is short. The author should include recent genome-wide studies such as: a. Genome-Wide Identification and Characterization of PIN-FORMED(PIN) Gene Family Reveals Role in Developmental and Various Stress Conditions in Triticum aestivum. b. Genome-wide identification and expression pattern analysis of the KCS gene family in barley. c. Genome-wide identification and characterization of abiotic stress-responsive lncRNAs in Capsicum annuum. d. Genome-Wide Identification and Characterization of the Brassinazole-resistant (BZR) Gene Family and Its Expression in the Various Developmental Stage and Stress Conditions in Wheat (Triticum aestivum). e. Genome-wide identification and functional characterization of natural antisense transcripts in Salvia miltiorrhiza. f. Genome-wide identification and expression analysis of the AT-hook Motif Nuclear Localized gene family in soybean. 2. What are the control in RT-qPCR analysis, and why is it not normalized? 3. Please make one hypothetical figure to explain the finding of this study. 4. Remove supplementary sequences from the main manuscript. I look hideous. Submit as supplementary material.

Reviewer 2 Report

The authors propose a manuscript titled “AvNAC030, a NAC Domain Transcription Factor and Enhances Salt Stress Tolerance in Kiwifruit”

The article is well structured. In particular, this study takes into consideration a topic aspect about suitable for neutral acid soil on Kiwifruit coltivation. The increasing of soil salinization, the adverse effects on the growth and development of plants and the declining yields and quality are discussed. The authors identified 120 NAC members and divided them into subfamilies according to phylogenetic analysis and analyse his systematic based on the conserved motifs, key amino acid residues in the domain, expression patterns, and protein interaction network predictions and screened the candidate gene AvNAC030. In order to study its function, the authors adopted the method of heterologous expression in Arabidopsis. The results show that the overexpression plants had higher osmotic adjustment ability and improved antioxidant defense mechanism. For the authors these results suggest that AvNAC030 plays a positive role in the salt tolerance regulation mechanism in kiwifruit.

I appreciate the original idea of the work which with a few revisions will convince me and the editor to publish it on Journal.

Abstract

Please introduce the reader also with the scientific name of Kiwi, Actinidia chinensis Planch, or other variety???

Introduction

  • Rows 40-41. Please follow my suggestion in order to complete the period: “Cultivating salt-tolerant crops, as well as the selection of salt-tolerant rootstocks of fruit trees (choose a reference), without excluding the chance to enhance the suitable native wild species (Perrino et al. 2021), are the most economical, effective, safe, and environmentally friendly methods

References do be added:

  • Perrino, E.V.; Valerio, F.; Jallali, S.; Trani, A.; Mezzapesa, G.N. Ecological and Biological Properties of Satureja cuneifolia Ten. and Thymus spinulosus Ten.: Two Wild Officinal Species of Conservation Concern in Apulia (Italy). A Preliminary Survey. Plants 2021, 10, 1952. https://doi.org/10.3390/plants10091952

  • Row 44. Please emphasize the richness in vitamin C, which is the antiviral vitamin par excellence and used as a cure for covid-19.
  • Please write the scientific plant name when cite for the first time kiwi that is Actinidia chinensis Planch;
  • Row 49. The Actinidia Genus… For botanical point of view when cite for the first time the species you must reporting also the author name.
  • Rows 51-52. Add reference for this statement “Kiwifruit is a fleshy root 51

that prefers neutral and acidic soil (choose reference)”.

  • Rows 80-84. After salt damage, kiwifruit morphology is mainly characterized by the inhibition of plant growth, the decline of organic matter accumulation, an insufficient supply of nutrients, short branches and internodes (choose a reference). The plant may stop growing or even die in serious cases. During the long-term evolution of plants, a set of resistance mechanisms against salt stress have been developed (choose a reference).
  • Row 103. See my previous comment: Arabidopsis
  1. Results

Well done the figures and tables are clear.

  • Only for Table S1 please check the size and font of column 1;
  • There are some grammar mistakes. Check!!!
  1. Discussion
  • Please two more words on vitamine C, see my previous comment on this tipic;
  • Rows 310-313. valvata Dunn, A. chinensis Planchon, A. deliciosa (Chev.) C. F. Liang & A. R. Ferguson, A. arguta (Siebold & Zucc.) Planch. ex Miq.
  • Rows 328 to 333. Please correct the scientific name of plant species
  • Brassica napus
  • Chrysanthemum lavandulifolium (Fisch. ex Trautv.) Makino
  • Chrysanthemum grandiflora Hook
  • Lilium pumilum Redouté
  1. Material and methods
  • Rows 409-410. Please give the geographical system used for geographic coodinates (WGS 84?)
  • If possible give a georeferenced map of place.

References

Please check and format in the correct way. Please consider DOI if available.

Reviewer 3 Report

In the manuscript entitled “AvNAC030, a NAC Domain Transcription Factor and Enhances

Salt Stress Tolerance in Kiwifruit” authors present the analysis of the NAC gene family in kiwifruit. There are many understatements and shortcomings in the presented manuscript. The introduction contains a lot of unnecessary repetition. I think that the manuscript is not suitable for publication in this form.

Title should be changed.

Line 17 – „key amino acid residues in the domain” – what domain?

Lines 68-71 – “Due to the lack of mineral nutrient elements necessary for growth, the formation and transportation of photosynthates are hindered, and the formation capacity of ATP and nutrients required for growth is reduced, resulting in nutrient deficiency and ion stress.”- this sentence should be rearranged.

Line 115 – “Subsequently, we verified the function of the gene” what gene?

The quality of the Fig. 1 should be improved.

Table S1 – names of the columns should be added

Line 170 – you are talking about the protein so it should not be written in italic

Lines 172-174 – “In addition, it has also been found that the C and D subdomains contain nuclear localization signals (NLSs), which may be related to the nuclear localization in transcription factors and the recognition of specific cis-acting elements on promoters.” - this sentence should be rearranged.

Lines 386-388 – “Duplicate and short sequences were manually deleted, and among similar sequences, leaving only members with the longest sequences.” - this sentence should be rearranged.

Table S3 should replaced to the supplementary materials.

Lines 397-398 – “We used Clustal Omega (http://www.ebi.ac.uk/Tools/msa/clustalo/) was used for phylogenetic analysis, which was then presented with the Interactive Tree of Life (iTOL) (https://itol.embl.de/itol.cgi).” - this sentence should be rearranged.

Lines 419-420 – “The N-terminal of AtBZR2 (AT1G19350.3) contained an NLS, which was fused

with mCherry as a nuclear marker” – I think this sentence is not finished, should be rearranged.

Table S5 – describe the purpose of using primers.

Procedure of cloning (4.4. Subcellular localization) should be more precisely described.

“4.5. The transformation of Arabidopsis and stress treatments” this section has to be rewritten since it is absolutely unclear. The method of RNA isolation? How the whole length of AvNAC030 was obtained.  Haw the full-length ORF of AvNAC030 was purified? Haw it was cloned into pCAMBIA3301 vector? I can only guess, I know this system, but it must be clearly presented to the reader.

“4.7. qRT-PCR analysis’ – which primers were used for Q-PCR? Was I the length of the obtained fragments?

Fig 10 – better description, what does VS, OE2……. mean.

What method was used for the relative expression analysis?

Lines  460-461 - “Using ZMH as the material, we performed high-throughput sequencing at the four time points after its salt treatment.” – There is no description in the manuscript, the authors only mentioned that this data are unpublished, so it should be described in the manuscript exactly.

Round 2

Reviewer 1 Report

I am happy with the author's reply and the manuscript can be accepted in its current form, However, there is odd sequencing in the manuscript. It should be removed before the acceptance of this manuscript.

Reviewer 2 Report

I appreciate the changes made by the authors, who have followed all the suggestions requested by me. In this latest version the manuscript is able to be published.

Regards,

Reviewer

Reviewer 3 Report

Lines 395-396 – In my opinion this sentence is still not proper  ” The repetitive and short sequences are removed, and the longest sequence is retained in the similar sequence”

We cannot use the term “Arabidopsis infection” – table S5

The authors answer do not satisfy me - “Using ZMH as the material, we performed high-throughput sequencing at the four time points after its salt treatment.” – There is no description in the manuscript, the authors only mentioned that this data are unpublished, so it should be described in the manuscript exactly. – In the revised manuscript there is no description about the method of library preparation, sequencing and analysis (Q 19 Lines  460-461 - “Using ZMH as the material, we performed high-throughput sequencing at the four time points after its salt treatment.” – There is no description in the manuscript, the authors only mentioned that this data are unpublished, so it should be described in the manuscript exactly. Thank you again for your valuable suggestions. It has been specifically described in Section 2.4.)

Round 3

Reviewer 3 Report

Accepted in this form